# Evaluating the accuracy of ChatGPT model versions for giving care-seeking advice
Marvin Kopka ⬦ ✉, Longqi He & Markus A. Feufel

## Abstract

**Background:** Artificial Intelligence tools such as ChatGPT are increasingly used by laypeople to support their care-seeking decisions, although the accuracy of newer models remains unclear. We aimed to evaluate the accuracy of care-seeking advice that is generated by all currently available ChatGPT models.

**Methods:** We evaluated 22 ChatGPT models using 45 validated vignettes, each prompted ten times (9,900 total assessments). Each model classified the vignettes as requiring emergency care, non-emergency care, or self-care. We evaluated accuracy against each case's gold standard solution (determined by two physicians), examined the variability across trials, and tested algorithms to aggregate multiple recommendations to improve accuracy.

**Results:** We show that o1-mini achieves the highest accuracy (74%), but we cannot observe an overall improvement with newer models – although reasoning models (e.g., o4-mini) improved their accuracy in identifying self-care cases. Selecting the lowest urgency level across multiple trials improves accuracy by 4 percentage points.

**Conclusions:** Although newer increasingly provide self-care advice, their accuracy remains insufficient for standalone use. However, making use of output variability with aggregation algorithms can improve the performance of existing models.

## Plain language summary

Many people use ChatGPT to decide whether they should go to the emergency room, see a doctor soon, or manage symptoms at home. We tested 22 model versions of ChatGPT with 45 real patient stories. Each story was asked ten times to test model consistency in suggesting one of three options: emergency care, non-emergency care, or self-care. The accuracy varied between all models; the best model gave correct recommendations in 74% of all cases. All models tended to advise more urgent care than needed, and they struggled most with self-care cases. Newer models were not generally better than older ones, although they gave correct self-care recommendations more often. When we combined a model's consecutive answers to the same query, accuracy improved slightly. ChatGPT may help people recognize emergencies, but it is not reliable enough to guide care on its own. Further safety testing is needed.

Artificial Intelligence (AI) applications are increasingly used in healthcare for tasks such as diagnosing patients, summarizing information, and answering health-related questions[1]. Although AI has traditionally been developed for such specific use cases, general-purpose foundation large language models (LLMs) – such as OpenAI's GPT or Anthropic's Claude models – have received increasing attention because they perform well across various domains and tasks[1]. For example, GPT-4 has passed several medical licensing exams, has proven effective in drafting letters and clinical notes, and appears to be useful in generating differential diagnosis lists with relatively high accuracy[2–10]. ChatGPT and similar large language models, however, are increasingly used by laypeople, because they are easy to access, easy to use, and applicable for a variety of different use cases[11–13]. For example, they are used for answering general health-related questions and

for diagnosing symptoms – both use cases in which ChatGPT has demonstrated moderate to high accuracy[12,14–16]. In terms of answering health-related questions, ChatGPT has been found not only to achieve accuracy levels comparable to medical experts but also to be perceived as more empathetic than medical professionals[17,18]. For self-diagnosis use cases, it has shown moderate accuracy compared to other self-diagnosis applications[7]. Given that self-diagnosis is not particularly useful for laypeople[19], using ChatGPT for obtaining care-seeking advice (which is also referred to as 'self-triage', 'triage advice', 'urgency advice' or 'dispositional advice') is said to be a more relevant use case for laypeople[19].

Applications for care-seeking advice, which are not based on LLMs, have been extensively tested and yielded mixed results. Some applications outperform laypeople and approach the performance of medical

Division of Ergonomics, Department of Psychology and Ergonomics (IPA), Technische Universität Berlin, Berlin, Germany.
✉e-mail: marvin.kopka@tu-berlin.de

professionals, whereas others perform worse than chance[20–22]. To date, few studies have benchmarked LLMs for care-seeking advice. Among those that have, most report a medium overall accuracy of about 70%, and a performance generally lower than that of medical professionals[7,16,23,24]. However, most of these studies have examined only a single LLM. Recent benchmarking tests from other domains suggest that newer models, such as GPT-4o and chain-of-thought (CoT) models (that are built on top of regular models and are automatically instructed to critically assess and revise their answers before giving output to simulate human meta-cognition[25]) like o1, perform better than their predecessors[10,26]. This suggests that ChatGPT models may also be improving their ability to provide care-seeking advice. However, no prior research has explicitly tested this hypothesis.

In addition to overall accuracy, output variability of LLMs across multiple trials using the same input should also be examined. Although output variability for the same input has been identified as a challenge when LLMs were used by professionals for diagnosing and treating patients[6–10], it has not yet been examined in the context of LLMs giving care-seeking advice. Prior studies have examined output variability only in non-LLM symptom-assessment applications or across prompts that use different examples[16,27]. If output variability for the same input exists, it may limit the interpretability of single-trial LLM evaluations. However, such variability could also be beneficial: research on 'the wisdom of the crowd' has shown that aggregating advice from multiple human advisors can improve the accuracy of decisions, and a similar approach may also improve the advice quality of LLMs[28]. To address these gaps, this study reports a benchmarking test with a longitudinal analysis to (a) compare the accuracy of care-seeking advice across all available ChatGPT models – as the most widely used LLM family[29] –, (b) analyze the variability of care-seeking advice across multiple trials, and (c) determine whether aggregating advice for the same case across multiple trials can be used to improve accuracy.

We find that average accuracy does not improve across 22 models with newer releases; the best model reached 74%. The models tend to overtriage and show moderate output variability, although reasoning models increase the accuracy among self-care cases. Aggregating multiple outputs of the same model by selecting the suggestion with the lowest urgency modestly improves overall accuracy and substantially improves self-care accuracy. We conclude that existing ChatGPT models may help recognize emergencies but are not reliable enough for standalone use, and that aggregating advice may be a strategy for developers to improve the accuracy of LLMs.

## Methods
### Study design
This study was designed as a longitudinal observational study comparing all available ChatGPT models. The primary outcome is the overall accuracy of the model's care-seeking advice. As secondary outcomes, we included additional accuracy metrics that have been proposed for evaluating applications for care-seeking advice in previous studies: the accuracy across different urgency levels (emergencies, non-emergencies, and self-care cases) and over- and undertriage errors (higher and lower urgency than appropriate)[30]. Also, we assessed the variability of the provided advice[31,32]. To determine how advice variability may be utilized to improve the accuracy of the model output, we also applied several pooling algorithms to aggregate the advice of the same model across multiple trials and assessed how they impact the accuracy measures identified above.

### Models
To capture trends in accuracy over time, all models released by OpenAI were eligible for inclusion. These models were identified through a document search of OpenAI's API documentation. The identified models can be categorized into two groups: generative pre-trained transformer (GPT) models and chain-of-thought (CoT) models. Because CoT models include a reasoning process, they take longer to produce an answer but are expected to improve response quality. Because each new model is expected to outperform its predecessors, we sorted all included models by release date.

### Case vignettes
We used a set of use-case specific case vignettes to evaluate the models, which was developed and validated in previous studies[24,33]. The set consists of real cases taken from individuals who described their symptoms in an online community to receive guidance and information on whether and where to seek care with their problems or symptoms. In total, 45 case vignettes were stratified in the set to reflect the natural base rate of both (1) emergency, non-emergency and self-care cases among laypeople using the internet for care-seeking advice, and (2) their symptom types according to the Centers for Disease Control and Prevention National Ambulatory Medical Care Survey coding protocol[24,34]. The chosen case vignette set thus has high external validity for generalizing to real-world scenarios in which laypeople use online decision support to determine whether and where to seek care for their health problems. However, given the naturally low frequency of emergencies, the set includes only two case vignettes describing emergency cases. The gold standard solutions for the cases were determined by two licensed physicians who independently rated the cases. In cases of disagreement, they discussed the case until reaching a consensus[24].

### Outcome evaluation
To assess the accuracy of care-seeking advice, each model was asked to classify each case vignette and assign the most appropriate triage urgency level. Following established evaluation methods for symptom checkers, the models were required to choose between three urgency options: emergency care, non-emergency care (including a visit to the family physician), or self-care (i.e., watchful waiting and not consulting a healthcare professional)[31,35]. To minimize input variability and potential bias and to increase comparability across studies, we used a standardized one-shot prompt to obtain responses. The prompt was developed in a previous study, which itself was adapted from two earlier studies[16,23,24]. The prompt reads:

*"Please give a triage classification for the following vignette. The criteria for triage classifications are presented below: Emergency: Call 911 or go directly to the emergency room; Non-Emergency: Try to see a doctor, but the symptoms do not require immediate attention at an emergency room. For example, you can see a doctor in the next week. Self-care: Let the health issue get better on its own and review the situation in a few days again. [Case Vignette]".*

To examine variability in advice, this prompt was used 10 times for each case description[10]. Model recommendations were obtained via the OpenRouter API using a custom-built Python script. This script added each case description to the prompt and then sent it to the OpenRouter API. To minimize bias, the context window was cleared before each API call, and the next case vignette was processed separately. Decoding parameters were set to default levels, that is, a temperature of 1.0, a top_p of 1.0, no max tokens, and no pre-specified seed value. The API requests were made from Berlin, Germany, on January 9, 2025. Because new models were released after the initial data collection, we conducted four additional rounds of API calls for these models on February 13, 2025 (for o3-mini and o3-mini-high), on April 23, 2025 (for o3, gpt-4.1-nano, gpt-4.1-mini, gpt-4.1, gpt-4.5-preview, o4-mini and o4-mini-high), on October 20, 2025 (for gpt-5 using both the non-reasoning and the standard reasoning setting), and on January 23, 2026 (for gpt-5.1 and gpt-5.2 using both the non-reasoning and the standard reasoning setting). The corresponding advice – which included the urgency level and background information or reasons for selecting this urgency level – was recorded and classified into 'emergency care', 'non-emergency care', or 'self-care' in a sequential procedure: first, two researchers manually classified a random selection of 50% of all cases. In cases of disagreement, both researchers re-evaluated the answer provided for the case and discussed it to determine the final classification. In a second step, we employed natural language processing (NLP) to automatically classify the models' output using gpt-4.1-nano and gpt-4.1-mini as cost-efficient but highly accurate models, each of which classified all outputs separately using a zero-shot prompt. We then assessed any disagreements between these two

models manually and compared their classification performance with the manual coding of all cases as the ground truth. Because this procedure yielded satisfying results, the other 50% of all cases were automatically classified using NLP and manual classification in case of disagreements between the two models.

The advice was classified as either correct (if it matched the gold standard solution) or incorrect (if it deviated from the gold standard solution). The primary outcome – overall accuracy – was calculated as the proportion of correctly classified cases across all repeated trials and urgency levels. Secondary outcomes were defined as follows: accuracy for each urgency level was calculated as the proportion of correctly classified cases within that specific urgency level. Overtriage errors were defined as cases in which the model's recommendation was of higher urgency than the gold standard solution, whereas undertriage errors were defined as cases in which the recommendation was of lower urgency then the gold standard solution. To assess each model's variability, Cohen's Kappa was calculated as a measure of agreement across ten trials for the same case. Additionally, to classify different levels of variability, advice for each model was categorized as follows: 'always correct' if every trial for a case yielded the correct result, 'sometimes correct' if some but not all trials contained the gold standard solution, or as 'never correct' if no single trial contained the gold standard solution.

To determine whether accuracy can be improved when using LLMs by aggregating multiple trials, we pooled the recommendations of all trials for each case and model and recalculated the accuracy (and accuracy across each urgency level). We applied four pooling algorithms: the first approach determined whether at least one of the 10 trials contained the gold standard solution. The second algorithm applied the majority rule by selecting the most common recommendation across all trials. The third algorithm used the highest urgency level that appeared in all trials for a case to overcome undertriage errors. The fourth pooling approach used the lowest urgency included in all trials for a case to overcome overtriage errors.

## Statistics and reproducibility

The data were analyzed in R using the symptomcheckR, DescTools, and tidyverse packages[32,36,37]. First, we calculated a two-way mixed, agreement, average-measures intra-class correlation (ICC) to assess inter-rater reliability between the two coders. We then determined the average accuracy and accuracy for each urgency level by calculating the proportion of correct responses. Hence, we treated correctness as a binary variable. Next, we used confusion matrices to visualize over- and undertriage errors, i.e., to assess the direction of disagreement between the models' recommendations and the gold standard solution. In this analysis, we treated the model's recommendations as an ordinal variable. Additionally, we used heatmaps to visualize the distribution of errors across all models. We then calculated Cohen's Kappa for each model to assess the degree of variability. Values below 0.40 were considered poor, between 0.40 and 0.54 weak, between 0.55 and 0.69 moderate, between 0.70 and 0.84 good and above 0.84 was considered excellent[38]. In the next step, we recalculated the accuracy after applying the pooling algorithms described above. Finally, we dichotomized the urgency levels to arrive at a binary choice (whether medical care is required, i.e., seek medical care or self-care) that laypeople commonly face, and calculated the sensitivity and specificity of each model[39,40].

## Results
### Models
Overall, 23 ChatGPT models existed at the point of the study. Given that GPT-3.5 has been discontinued, we included 22 models in our analysis. Each model was prompted in 10 trials with all 45 cases, resulting in an assessment

**Fig. 1 | Accuracy of all ChatGPT models sorted by release date (from left to right) across different urgency levels.** The error bars indicate the 95% confidence interval of the mean estimate. For each model, n = 45 independent cases were included to obtain the overall mean accuracy estimate, n = 2 to obtain emergency accuracy, n = 30 to obtain non-emergency accuracy, and n = 13 to obtain self-care accuracy. Emergency accuracy may be unreliable because the vignette set contained only two emergency cases.

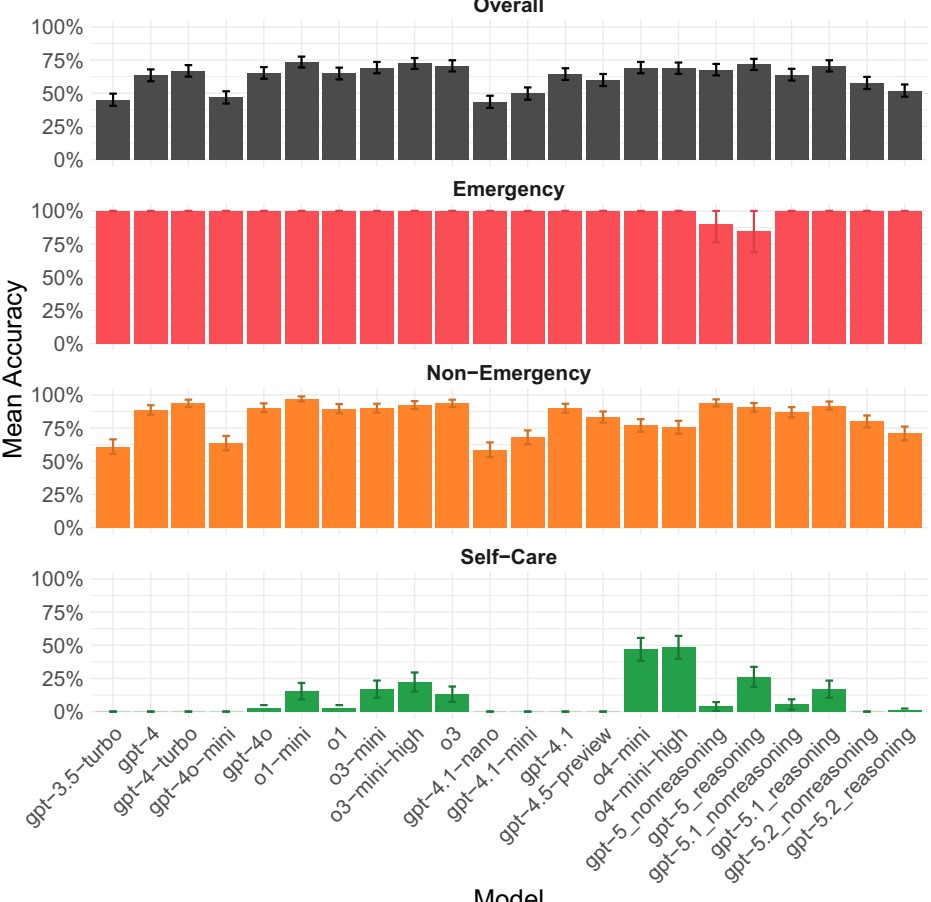

of 9900 cases. Both the inter-rater agreement of the coders classifying the model output manually and the inter-rater agreement of the two NLP models were very high (ICC = 0.997 and ICC = 0.996, respectively). After manually assessing and coding disagreements between the NLP models (31/9,900 cases), the classification models classified all cases (100%) correctly according to the raters' classification ground truth.

## Accuracy

On average, o1-mini demonstrated the highest accuracy with 74% of cases solved correctly (331/450, [95% CI 69–78%]), whereas gpt-4.1-nano had the lowest accuracy with 44% of cases solved correctly (196/450, [95% CI 39–48%]). Although all models except gpt-5 correctly identified both emergency cases in all trials (with both gpt-5 non-reasoning and reasoning correctly identifying emergencies in 18/20 and 17/20 of all trials respectively), o1-mini had the highest accuracy in correctly classifying non-emergency cases (97%, 291/300, [95% CI 95–99%] and o4-mini-high the highest accuracy for self-care cases (48%, 63/130, [95% CI 40–57%]), see Fig. 1.

All models had the tendency to assign a higher level of urgency than the gold standard solution. This tendency is particularly evident for gpt-3.5-turbo, where all cases were assigned an urgency level that was either correct or higher than necessary. In newer models, the tendency to overtriage first decreased and then increased again in gpt-5. Specifically, the proportion of overtriage errors among all errors decreased from 100% (247/247, [95% CI 100–100%]) for gpt-3.5-turbo to 69% (96/140, [95% CI 60–76%]) for o4-mini-high and increased again to 99% (216/217, [95% CI 97–100%] for gpt-5.2_reasoning), see Fig. 2.

## Errors

Most errors (70%, 2573/3696, [95% CI 68–71%]) were observed in self-care cases. No self-care case was solved correctly by all models across all trials (0%, 0/13, [95% CI 0-23%]); 10/13 (77%, [95% CI 46–95%]) were solved correctly at least once, and 3/13 (23%, [95% CI 8-50%]) were never solved by any model. Item difficulty (proportions solved for each case) for self-care cases ranged from 0% to 27%. For non-emergency cases, two cases (7%, 2/30, [95% CI 1–22%]) were solved correctly by all models in all trials, all (100%, 30/30, [95% CI 88-100%]) were solved correctly at least once, and none remained unsolved (0%, 0/30, [95% CI 0-12%]). Item difficulty ranged from 41% to 100%. Among the two emergency cases, one (50%, 1/2, [95% CI 1–99%]) was solved correctly by all models in all trials; both (100%, 2/2, [95% CI 16–100%]) were solved correctly at least once, and none remained unsolved (0%, 0/2, [95% CI 0–84%]), see Fig. 3. Item difficulty ranged from 97% to 100%.

## Output variability

Outputs for the same case varied across multiple trials with no clear trend over time. However, some models (such as gpt-4.1 and gpt-4.5-preview) demonstrated very low variability. Cohen's Kappa was moderate to high for all models, see Table 1. Whereas older models had a high proportion of cases that were never solved correctly (49%, 22/45, [95% CI 34–64%] for gpt-3.5-turbo), newer models had a lower proportion (9%, 4/45, [95% CI 2–21%] for o4-mini-high). However, newer models had a higher proportion of cases that received inconsistent advice across multiple trials, that is, that were solved correctly in some trials and incorrectly in other trials (44.4%, 20/45, [95% CI 30–60%] for o4-mini-high and 11%, 5/45, [95% CI 3-24%] for gpt-3.5-turbo), see Fig. 4. Notably, only one model (gpt-4.5-preview) was either always or never correct, without any variability.

## Accuracy improvement with pooling

Results of the pooling rules are displayed in Fig. 5. Not surprisingly, performance across all models improved the most compared to their overall accuracy by coding a correct solution once the model solved a case correctly at least once across all ten trials ($M_{all\_models}$ = +9 percentage points, ranging from 0 to 22 percentage points). The second highest performance

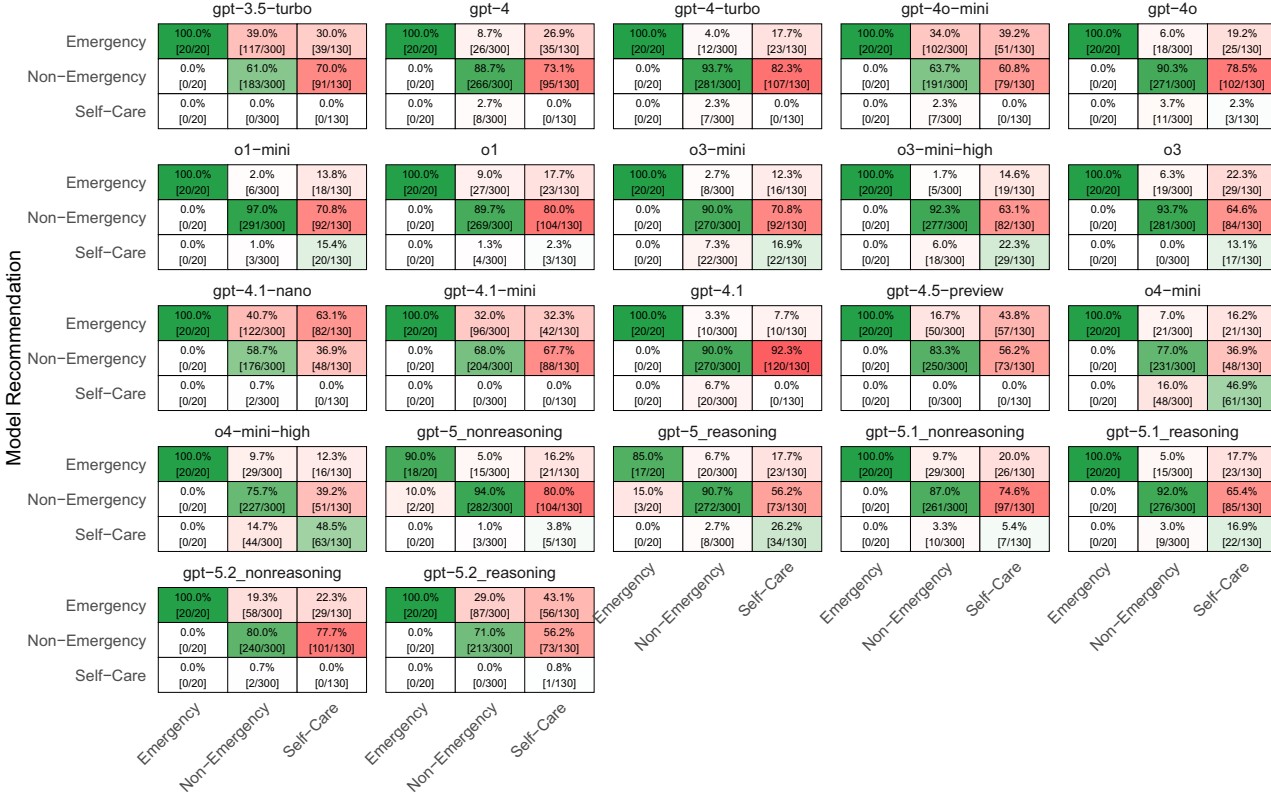

**Fig. 2 | Confusion matrix of urgency levels based on gold standard solutions (x-axis) versus ChatGPT model recommendations (y-axis).** Incorrect recommendations are shown in red, whereas correct recommendations are green.

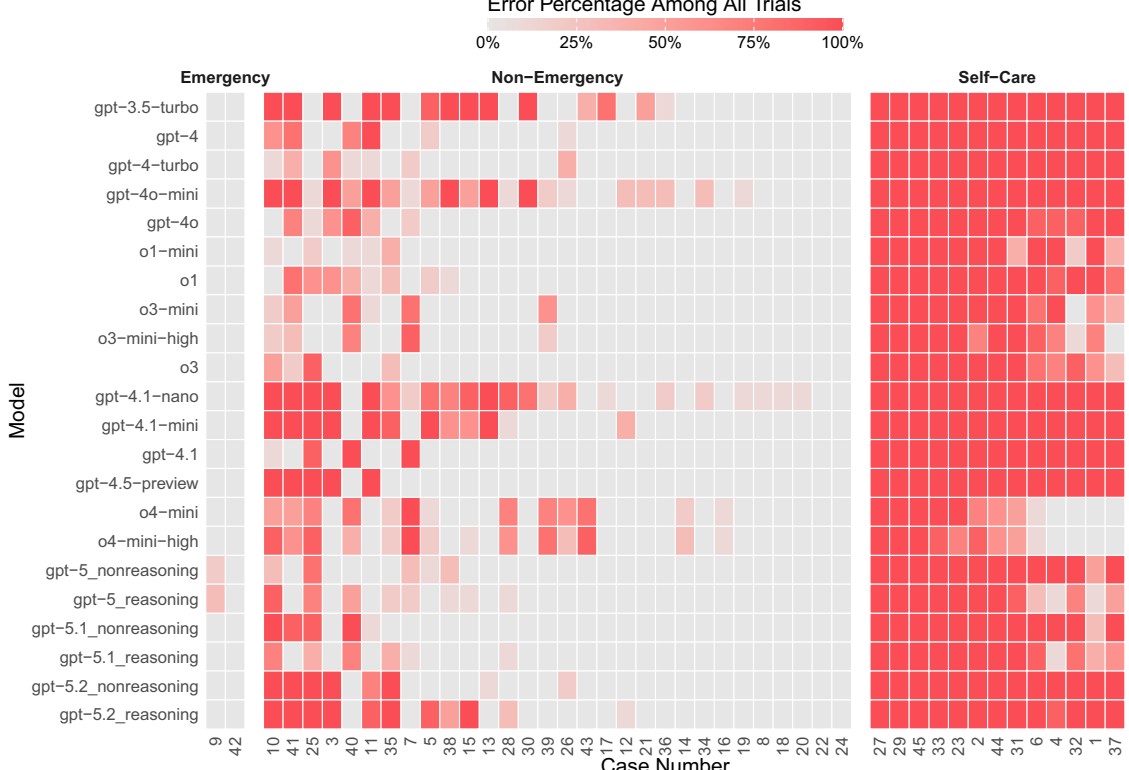

**Fig. 3 | Heatmap showing the error percentage of each model among all trials for each case, sorted by the proportion of non-solved cases (from left to right) within each urgency level.** Each cell contains $n = 10$ trials to obtain the error percentage of each model for each case. The intensity of red color shows how often the corresponding case has *not* been solved by the corresponding model across all ten trials.

**Table 1 | Interrater agreement of all ChatGPT models across ten trials**

| Model | Cohen's Kappa |
|---|---|
| gpt-3.5-turbo | 0.84 |
| gpt-4 | 0.81 |
| gpt-4-turbo | 0.68 |
| gpt-4o | 0.71 |
| gpt-4o-mini | 0.67 |
| o1-mini | 0.69 |
| o1 | 0.67 |
| o3-mini | 0.71 |
| o3-mini-high | 0.75 |
| o3 | 0.71 |
| gpt-4.1-nano | 0.67 |
| gpt-4.1-mini | 0.82 |
| gpt-4.1 | 0.96 |
| gpt-4.5-preview | 0.97 |
| o4-mini | 0.71 |
| o4-mini-high | 0.70 |
| gpt-5_nonreasoning | 0.68 |
| gpt-5_reasoning | 0.65 |
| gpt-5.1_nonreasoning | 0.89 |
| gpt-5.1_reasoning | 0.69 |
| gpt-5.2_nonreasoning | 0.93 |
| gpt-5.2_reasoning | 0.85 |

improvement across all models was achieved when coding the lowest urgency advice across all trials as the final advice ($M_{all\_models} = +4$ percentage points, ranging from $-0.2$ to 12 percentage points). Following the majority rule resulted in a decreased performance ($-0.2$ percentage points, ranging from $-5$ to 4 percentage points), as well as using the highest urgency advice across all trials ($M_{all\_models} = -6$ percentage points, ranging from $-19$ to 0 points). Notably, self-care accuracy improved substantially – albeit not all models gave self-care advice – when using the lowest urgency across all trials ($M_{all\_models} = 14$ percentage points, ranging from 0 to 28 percentage points). For example, o4-mini-high could solve 77% (10/13, [95% CI 46-95%]) of self-care cases correctly using this pooling algorithm, whereas taking the mean across all trials resulted in a self-care accuracy of 48% (63/130, [95% CI 40-57%]), see Fig. 5.

### Performance in binary choice

When dichotomizing the urgency levels to represent decisions between medical care and self-care, gpt-3.5-turbo, o3, gpt-4.1-mini, gpt-4.5-preview, and gpt-5.2_reasoning showed the highest sensitivity (100%), and o4-mini the lowest (85%). The model o4-mini-high showed the highest specificity (49%), and gpt-3.5-turbo, gpt-4, gpt-4-turbo, gpt-4o-mini, gpt-4.1-nano, gpt-4.1-mini, gpt-4.1, gpt-4.5-preview, and gpt-5.2_nonreasoning the lowest (0%), see Table 2.

### Discussion

In this study, we benchmarked all available ChatGPT models to assess the accuracy of the care-seeking advice they provide using real cases. Our results show that average accuracy remains relatively constant across models and does not improve with newer iterations. However, self-care accuracy has improved over time, particularly with the introduction of CoT models and o4-mini. It should be noted, however, that variability strongly depended on the specific model used. To account for this variability, we explored pooling

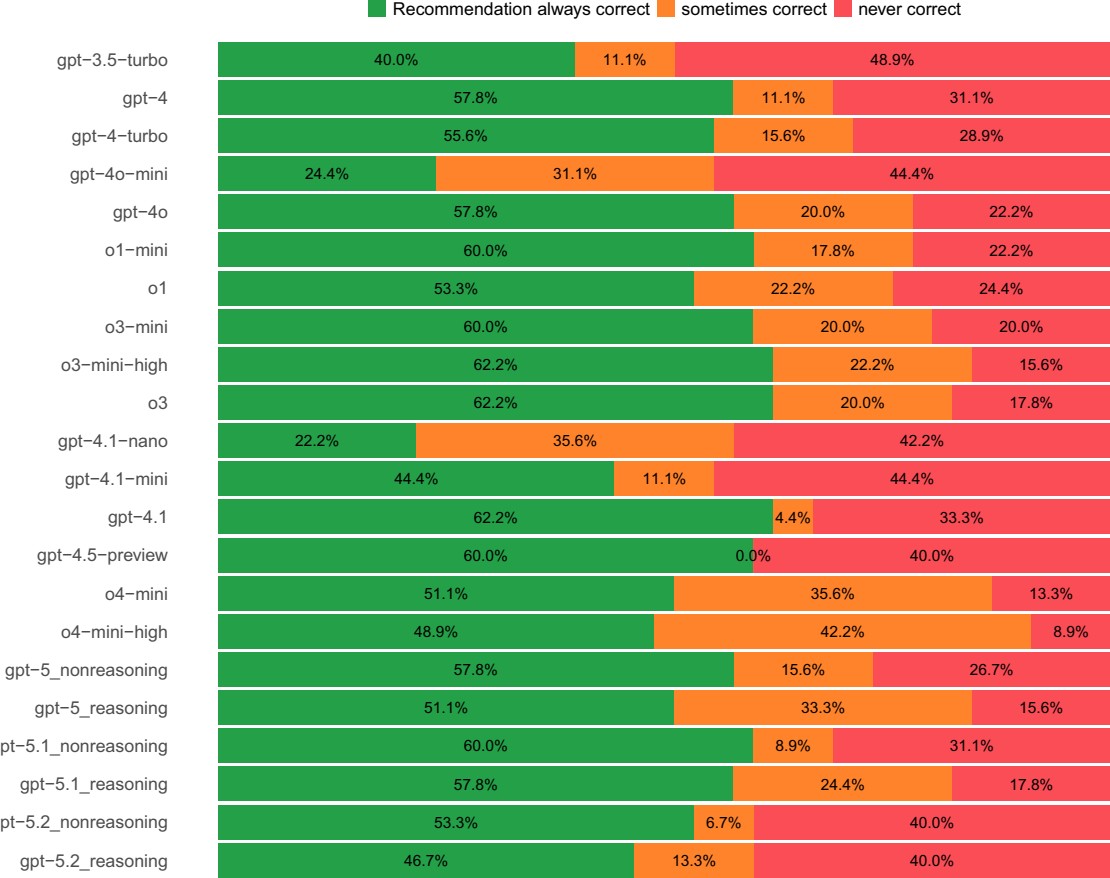

**Fig. 4 | Model variability/consistency of recommendations per case, for 45 cases and ten trials.** The values indicate how often all ten trials yielded the same recommendation that was either correct (green) or incorrect (red) for the same case or whether recommendations differed within ten trials (orange).

the recommendations for the same case across multiple trials using different algorithms. Our results indicate that – because the models appear to be risk-averse – using the lowest urgency recommendation may be an effective strategy to increase the accuracy of LLMs. These findings are discussed next.

In general, our accuracy estimates align with those reported in previous studies and tested with various vignette sets. For example, a previous study using the same case vignettes reported an accuracy of 71% for GPT-4, and another study using different case vignettes reported an accuracy of 67%[23,24]. A third study, which tested both gpt-3.5 and gpt-4 with a third set of case vignettes, reported an accuracy of 59% for gpt-3.5 and 76% for gpt-4, respectively[7]. In other words, existing estimates match our findings for the same models, even though all studies used different case vignettes, which we interpret as evidence for the validity of the existing data and our results.

Our study extends previous research by including and comparing a larger number of models. Results show that the average accuracy of all ChatGPT models remains relatively stable at approximately 70%, with only the oldest model, gpt-3.5-turbo, and smaller models like gpt-4o-mini, gpt-4.1-nano, and gpt-4.1-mini performing markedly worse. Despite these deviations, we did not observe any significant improvement in average accuracy across newer models. However, we observed that self-care recommendations are more readily provided with the advent of CoT models. In a previous study, Levine et al. hypothesized that the low proportion of self-care recommendations might be due either to a lack of self-care discussions in the online data used to train the models or to a need for common-sense reasoning to arrive at a self-care decision[16]. Given that our self-care cases were identified online, and because self-care recommendations increased with CoT models – which include a reasoning process before generating recommendations – our findings suggest that increased common-sense reasoning may be the driver. In fact, one explanation for the

performance increase in CoT models is that their reasoning approach may circumvent safety measures that previously prevented models from providing self-care advice to minimize potential harm to users.

Despite the increase in self-care recommendations, the accuracy of identifying self-care cases remains far from perfect and currently offers limited decision support for users. This can be explained by the models' conservative triage behavior: all of them have a tendency to classify cases as requiring medical care, as evidenced by high sensitivity (85% to 100%) and low specificity (0% to 49%) across models. From a clinical perspective, some degree of overtriage can be preferable to undertriage because the potential harm of recommending unnecessary care is typically smaller than the harm of missing an emergency. However, the problem is how conservative the models are. If a model rarely or never recommends self-care, its advice is effectively to seek medical care in all cases, which limits its value as decision support. At a systems level, these recommendations may also increase avoidable healthcare utilization and costs and may indirectly worsen outcomes for other patients if they contribute to overcrowding in emergency departments[41,42]. At an individual level, always advising medical care for minor symptoms may increase anxiety and reassurance seeking in vulnerable users (e.g., those with health anxiety or tendencies to obsessive-compulsive behavior)[43]. Over time, users may also learn an unhelpful heuristic (that any single minor symptom warrants medical care), which further exacerbates these negative effects.

An additional factor limiting the usefulness of self-care advice by existing ChatGPT models is the substantial variability in the outputs. Because self-care is only occasionally recommended across multiple trials, even if it is the correct solution, users are likely to be advised to seek medical care more often than necessary. Given that laypeople generally recognize emergency cases reliably but have problems determining whether self-care is an appropriate option[24,44], future models and applications that build on these

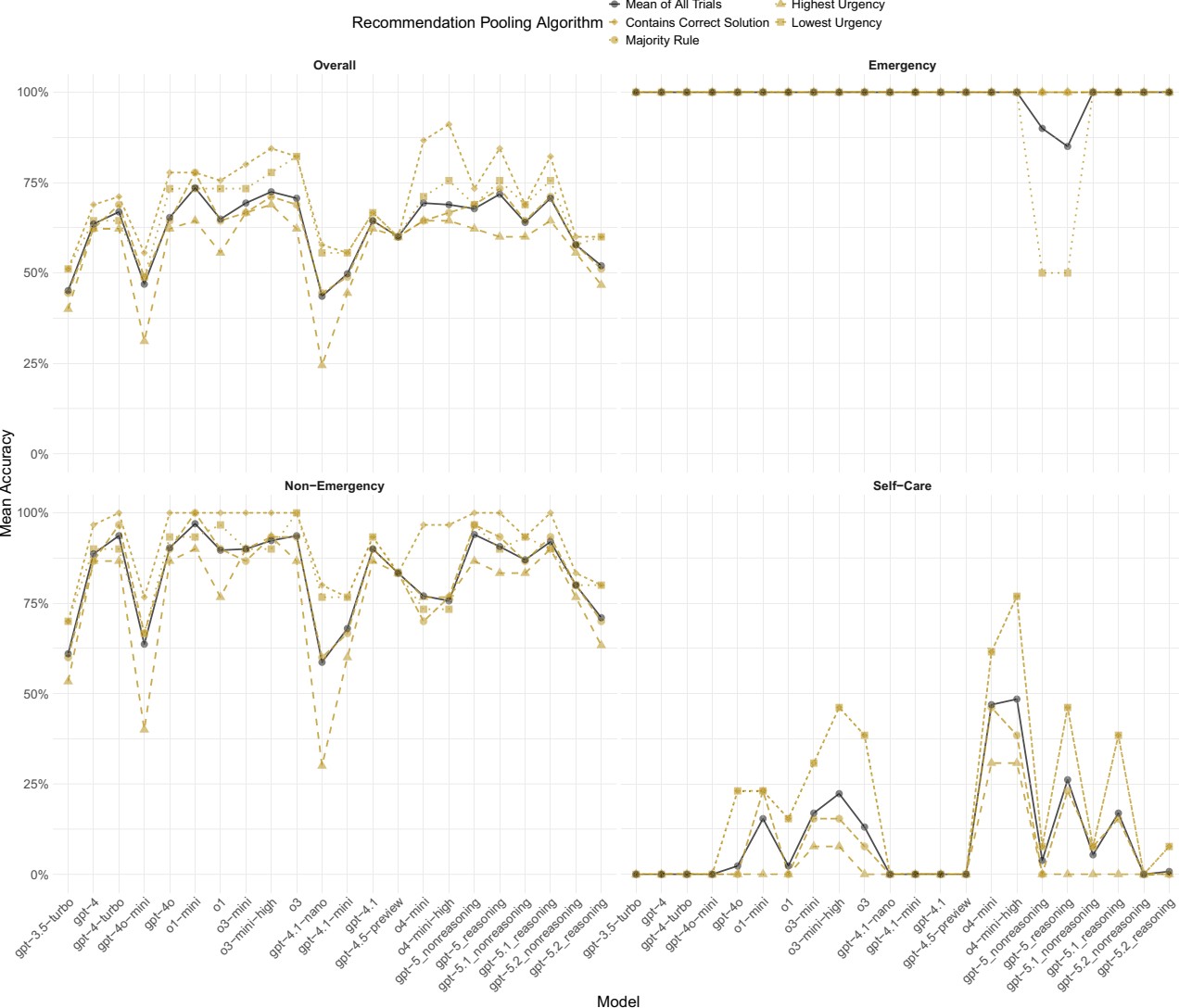

**Fig. 5 | Accuracy of all ChatGPT models when recommendations are pooled according to various algorithms.** For each model, $n = 45$ independent cases were included to obtain the overall mean accuracy estimate, $n = 2$ to obtain emergency accuracy, $n = 30$ to obtain non-emergency accuracy, and $n = 13$ to obtain self-care accuracy. Emergency accuracy may be unreliable because the vignette set contained only two emergency cases.

models should strive to further improve the accuracy of the self-care advice if their goal is to assist in care-seeking decisions.

Our analyses suggest that a potential strategy to further increase (self-care) accuracy could involve utilizing output variability, i.e., prompting the model multiple times and aggregating its recommendations. In our controlled setting, selecting the lowest urgency across multiple trials increased self-care accuracy without a large decrease in other urgency levels. However, this approach should not be viewed as a deployment recommendation for end-users, as it could miss emergencies with fatal consequences. Rather, CoT models may apply a pooling algorithm similar to ours to improve the care-seeking advice it provides, with additional clinical validation to ensure that the provided advice is safe for users.

This study has several implications for practice and regulation. First, the current and previous findings indicate that currently available ChatGPT models may be effective for identifying emergencies[16,23]. Laypeople may therefore use them and critically assess recommendations if they are uncertain whether emergency care is necessary, although they are not sufficient for standalone use; when in doubt, users should follow local emergency guidance. For cases other than emergencies, most ChatGPT models are currently not reliable tools, in particular if users are unsure whether they require medical care or can rely on self-care and stay at home. Only o4-mini

and o4-mini-high outperformed laypeople, although they still correctly identified only half of all self-care cases[24]. However, prompting these models multiple times and selecting the lowest urgency recommendation can substantially increase self-care accuracy, with only minor reductions in correctly identifying emergency and non-emergency cases. Because of potential safety concerns, we recommend that this strategy be limited to developers with additional safety testing and not used by end-users. For end-users, it may be reasonable to ask the same question multiple times, and – if possible – across different models (e.g., o4-mini and o4-mini-high, which showed higher self-care accuracy than laypeople) and to critically assess and aggregate the recommendations they receive.

This study also demonstrates that AI should be regulated and evaluated according to specific use cases. Because ChatGPT is a general-purpose tool and not marketed as providing medical advice, it is generally not regulated as a medical device[45]. When it is used privately for health information without official medical claims, consumer and data-protection laws apply rather than medical-device rules[45,46]. Hence, it is important to clarify how future regulation can mandate the safety of ChatGPT and other LLMs, which use cases should be regulated, and whether this should apply only in clinical settings or also to private use. As our results demonstrate, although ChatGPT models seem to have high utility in some use cases such as text

**Table 2 | Sensitivity, specificity, and F1 score of all ChatGPT models**

| Model | Sensitivity | Specificity |
|---|---|---|
| gpt-3.5-turbo | 100% | 0% |
| gpt-4 | 98% | 0% |
| gpt-4-turbo | 98% | 0% |
| gpt-4o | 97% | 2% |
| gpt-4o-mini | 98% | 0% |
| o1-mini | 99% | 15% |
| o1 | 99% | 2% |
| o3-mini | 93% | 17% |
| o3-mini-high | 94% | 22% |
| o3 | 100% | 13% |
| gpt-4.1-nano | 99% | 0% |
| gpt-4.1-mini | 100% | 0% |
| gpt-4.1 | 94% | 0% |
| gpt-4.5-preview | 100% | 0% |
| o4-mini | 85% | 47% |
| o4-mini-high | 86% | 49% |
| gpt-5_nonreasoning | 99% | 4% |
| gpt-5_reasoning | 98% | 26% |
| gpt-5.1_nonreasoning | 97% | 5% |
| gpt-5.1_reasoning | 97% | 17% |
| gpt-5.2_nonreasoning | 99% | 0% |
| gpt-5.2_reasoning | 100% | 1% |

summarization, self-diagnosis, and answering general medical questions, the usefulness of most models in terms of care-seeking advice is limited[4–9,12,14,15]. In fact, by frequently recommending a higher urgency level than medically necessary, these models may actively contribute to increased and potentially inappropriate healthcare utilization, with the current economic impact of mis-guided resource utilization estimated at over four billion dollars annually already[41,47]. This risk is further exacerbated by high variability in outputs and the fact that self-care was rarely advised across multiple trials, even when it was the most appropriate recommendation. Future evaluation methods and regulations should thus explicitly specify the use case AI tools are tested for and should also find ways to deal with variability in LLM output. They should also test multiple models and identify those that perform best in a selected use case. As our findings demonstrate, accuracy varies depending on how multiple outputs for the identical prompt are pooled. If evaluators rely solely on 'technical accuracy' (i.e., whether the correct solution was provided at least once), they risk overestimating the model's real-world accuracy. Instead, to obtain a more accurate estimate of its real-world impact, using the mean of all recommendations may be a more effective approach, as it better reflects the advice (and variability) received by an average user.

Our study has several limitations. First, the prompt used in this evaluation is artificial, and laypeople are unlikely to phrase their queries in the same way it was designed for this study. In real-world settings, models may also ask follow-up questions to clarify ambiguous inputs, and users can provide additional information – both of which can affect LLM accuracy. However, prompting strategies vary widely among users, and the exact wording depends entirely on the individual using it[44]. We adopted a prompt used in previous studies to ensure cross-study comparability and to reduce potential bias because of input variability of different users[16,23,24]. Additionally, we used a standardized and validated vignette set and benchmarking methodology to assess the accuracy of all models[24]. If future studies want to evaluate the impact in real-world scenarios, i.e., how ChatGPT influence users' decisions, they should consider observational designs or

clinical studies involving participants who generate their own prompts based on their own symptoms. Future studies may also explore different prompting strategies that laypeople use and how they affect the LLMs' accuracy.

The second limitation concerns the gold standard solutions to the case vignettes, which were determined by two physicians but represents only an approximation to the ground truth. Establishing the ground truth would require a comprehensive clinical assessment, including lab tests, imaging, and long-term follow-up, to accurately determine the cause of symptoms and thus the most appropriate urgency level[48]. However, we followed established guidelines in developing the gold standard solution, and our findings align with those of other studies in which urgency ratings were provided by different experts[16,23].

The last limitation concerns the distribution of urgency levels in our dataset. The case set was developed to reflect natural base rates of all urgency levels, but true medical emergencies among users of care-seeking decision aids in the real world are relatively rare. This led to average accuracy values that are closer to real-world performance but introduces class-imbalance, as only two emergency cases were included. This imbalance makes estimates for emergency cases imprecise and may overstate accuracy for that urgency level, which limits generalizability. Although studies with more emergency cases have reported consistently high accuracy in identifying emergencies as well, our estimate for this urgency level should be interpreted cautiously[16,23]. Future studies should replicate our study with a larger dataset that includes more emergency cases, but is also stratified to reflect natural base rates across emergency, non-emergency cases, and self-care cases.

## Conclusions

Our study demonstrates that although the overall accuracy of ChatGPT models in providing care-seeking advice does not improve, these models have become better at identifying cases where self-care is appropriate––but certainly not sufficiently. Because self-care is the most valuable form of advice for users, the current utility of most ChatGPT models as decision support systems for care-seeking decisions remains limited and only o4-mini and o4-mini-high with an additional advice-pooling algorithm may be useful. Prompting different models multiple times and aggregating recommendation can increase their accuracy in identifying self-care cases. Because there is currently no one-size-fits-all model––some models perform better with more urgent cases, while others perform better with low-urgency cases––users should ideally test, critically assess, and aggregate recommendations from multiple ChatGPT models or other non-LLM applications rather than relying on a single general-purpose model.

## Data availability

The source data for all figures and analyses can be found in Supplementary Data 1.

## Code availability

We used Python with pandas 2.2.3, requests 2.32.3, and json5 0.10.0 to collect the data. We used R and tidyverse 2.0.0, irr 0.84.1, and symptomcheckR 0.1.3 to analyze the data. The code is available from the corresponding author upon reasonable request.

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

## Author contributions

M.K. and M.A.F. conceived of the study. M.K. developed the procedure. M.K. and L.H. collected the data. M.K. conducted the analyses and data visualization and wrote the first draft of the manuscript. All authors provided critical input and worked on manuscript development.

## Funding

## Competing interests

The authors declare no competing interests.
