## [Transparent Peer Review file · Communications Medicine]

Evaluating the Accuracy of ChatGPT Model Versions for Giving Care Seeking Advice

Corresponding Author: Dr Marvin Kopka

Version 0:

Reviewer comments:

Reviewer #1

(Remarks to the Author)

This paper addresses an interesting and timely topic, by evaluating 16 different ChatGPT models on medical case vignettes using accuracy as the main metric. It's great to see a broad comparison like this, especially since these tools are increasingly being used in health-related contexts. That said, there are a few areas where the paper could be improved or expanded to make the findings more robust and relevant.

- 1) It would be helpful to include more performance metrics beyond accuracy. In a medical context, measures like sensitivity, specificity, or F1-score are often more informative, especially when the dataset is small and imbalanced.
- 2) The small size and imbalance of the dataset make interpretation challenging. Although this is already partially done: it would be good to acknowledge this more clearly and possibly discuss how it limits the generalizability of the results.
- 3) The suggested aggregation strategy—taking the lowest rating across model outputs—might be questionable in this medical context. While conservative decision-making can make sense in medicine, relying solely on the “lowest” answer might be overly simplistic in a triage setting, especially in emergency scenarios where even minor performance drops could have serious consequences.
- 4) A point to consider is the discussion around regulation. While I agree that regulation is important when it comes to AI in healthcare, I think the argument needs more nuance. These models aren't being used as medical tools per se. They're being used privately by individuals, often outside any clinical setting. That makes regulation both more complex and, in some ways, harder to enforce. A clearer distinction here would strengthen the paper's argument.
- 5) Some exploration of prompt engineering would be insightful. It would be valuable to see how the different models react to varied prompts or less clearly formulated questions. That would give a better picture of real-world performance, where users don't always phrase things perfectly.
- 6) It's worth mentioning that in a real-world scenario, models can ask follow-up questions to clarify ambiguous inputs. That kind of interactive exchange is an important part of how these tools are used but isn't covered in this study.
- 7) Some analysis of the errors themselves would be great to better understand the model differences. Are the same cases misclassified across all models?
- 8) I found the conclusion that users should choose the model based on their case a bit unrealistic. Expecting from users to assess the model's suitability case by case feels disconnected from how these tools are actually used and how much most of the users know about the underlying technology.

Reviewer #2

(Remarks to the Author)

The authors benchmark 16 ChatGPT models on 45 validated care-seeking vignettes, each prompted 10 times (7200 assessments). They report overall accuracy (~70%) remains stable across model generations, with modest improvements in

self-care classification among chain-of-thought (CoT, "reasoning") models. They also explore across-trial variability and demonstrate that pooling strategies, particularly selecting the lowest urgency across trials, can increase accuracy, especially for self-care. The study concludes that while ChatGPT models outperform laypeople in some respects, their accuracy remains insufficient for standalone clinical use.

The manuscript addresses an important, practical question and brings several innovations (breadth of model benchmarking, aggregation strategies). However, the numerical inconsistencies, emergency case scarcity, unclear variability narrative, incomplete reproducibility details, and some overstatements of practical implications are significant issues. Also, since GPT-5 is now publicly available, incorporating it into the benchmarking would substantially improve the timeliness and relevance of the manuscript.

Major Strengths

- Breadth and longitudinal perspective: The inclusion of essentially all available ChatGPT models, tested under a consistent protocol, adds valuable longitudinal context beyond single-model snapshots.
- Attention to reproducibility: The study carefully describes prompting procedures, context clearing, and dates of model access, enhancing transparency.
- Focus on variability and aggregation: Evaluating across-trial instability and testing pooling rules is both novel and practical, and contributes to understanding how real-world users might experience LLM advice.
- External validity: Vignette set is previously validated, stratified by base rates, and grounded in real patient cases.

Major Concerns

1. Several accuracy values in text and figures do not match the reported numerators/denominators - like:

- o1-mini...78% (331/450), but $331/450 = 73.6\%$.

- o4-mini...22% (69/130), but $69/130 = 53.1\%$.

However the confidence intervals align with the corrected point estimate, suggesting typographic or calculation errors.

Regardless, these need systematic reconciliation before the results can be interpreted with confidence.

2. Only two emergency cases were included. Is that a limitation of the validated set? Regardless, claims such as "all models correctly identified all emergencies" should be softened, as accuracy estimates and CIs are unreliable with $n=2$. The authors should either expand the emergency case set or temper conclusions about this class.

3. The text claims variability "tends to increase with newer models," but Table 1 shows excellent kappas for gpt-4.1/4.5-preview (0.96–0.97). Figure 3 suggests more nuanced patterns. The narrative should be aligned with the actual variability measures.

4. Variability is highly sensitive to decoding parameters. The manuscript does not report temperature, top_p, max_tokens, or seed values used in OpenRouter calls. Without these, reproducibility is limited. Please include full parameterization and any relevant system prompts.

5. The authors use GPT to classify GPT. Output classification relied in part on gpt-4.1-nano/mini, with human adjudication of disagreements. Because those same model families are under evaluation, correlated errors are possible. A sensitivity analysis using only human labels on a subset would strengthen confidence that findings are not biased by automated adjudication.

6. The introduction implies that output variability "has not yet been examined" for care-seeking advice, but prior work by Fraser for WebMD symptom checkers is arguably such an analysis. Levine (16) too. Similarly, define and cite "CoT models" where first introduced.

7. The Discussion suggests laypeople should use ChatGPT to decide about emergencies. Given safety, regulatory, and liability concerns, and the very limited emergency sample, this should be reframed more cautiously, like "may assist but is not sufficient for standalone use; when in doubt, follow local emergency guidance."

8. Since GPT-5 is now publicly available and many older models are deprecated, the authors should strongly consider incorporating it into their analysis, as it would substantially improve the timeliness and relevance of the manuscript.

Minor Comments

- Clarify terminology: The text alternates between "laypeople" and "patients" as if disjoint; define terms consistently.
- Abstract and Results numbers must match exactly.
- "In addition to overall accuracy, also output variability..." (smoother phrasing needed)
- Low-resolution figures (especially axis labels in Figures 1–4) should be improved.
- Consider placing emergency vignette results in Supplementary material with caveats in the main text.

Statistical Analysis and Reproducibility: The analytic framework (proportions, confusion matrices, Cohen's kappa, ICCs) is appropriate. However, the numerical inconsistencies undermine confidence in the reported estimates. Reproducibility is limited by omission of key decoding parameters and reliance on LLM adjudication without full human replication.

Version 1:

Reviewer comments:

Reviewer #1

(Remarks to the Author)

The authors have satisfactorily addressed all previous comments. I have no further comments and recommend acceptance.

We would like to thank both reviewers for the excellent feedback, which helped us improve the quality of our manuscript substantially. Please find our responses below.

Reviewer #1:

This paper addresses an interesting and timely topic, by evaluating 16 different ChatGPT models on medical case vignettes using accuracy as the main metric. It's great to see a broad comparison like this, especially since these tools are increasingly being used in health-related contexts. That said, there are a few areas where the paper could be improved or expanded to make the findings more robust and relevant.

1) It would be helpful to include more performance metrics beyond accuracy. In a medical context, measures like sensitivity, specificity, or F1-score are often more informative, especially when the dataset is small and imbalanced.

Response: Thanks for this suggestion. We originally decided against including such scores, because they require a binary decision and we report results of a three-tiered decision. However, after re-consideration, we decided to conduct an additional analysis where we dichotomized the three tiers and calculated the scores for a decision scenario where the question is whether to seek care or not. This decision represents a common decision that medical laypeople face. We now report sensitivity, specificity, and the F1-score for this decision at the end of our results.

2) The small size and imbalance of the dataset make interpretation challenging. Although this is already partially done: it would be good to acknowledge this more clearly and possibly discuss how it limits the generalizability of the results.

Response: We acknowledged this limitation more clearly and added a brief discussion of this limitation to the manuscript: "The case set was developed to reflect natural base rates of all urgency levels, but true medical emergencies among users of care-seeking decision aids in the real world are relatively rare. This led to average accuracy values that are closer to real-world performance but introduces class-imbalance, as only two emergency cases were included. This imbalance makes estimates for emergency cases imprecise and may overstate accuracy for that urgency level, which limits generalizability. Although studies with more emergency cases have reported consistently high accuracy in identifying emergencies as well, our estimate for this urgency level should be interpreted cautiously.^{16,23} Future studies should replicate our study with a larger dataset that includes more emergency cases but is also stratified to reflect natural base rates across emergency, non-emergency cases, and self-care cases."

3) The suggested aggregation strategy—taking the lowest rating across model outputs—might be questionable in this medical context. While conservative decision-making can make sense in medicine, relying solely on the "lowest" answer might be overly simplistic in a triage setting, especially in emergency scenarios where even minor performance drops could have serious consequences.

Response: Thanks for this comment. We agree that this recommendation was over simplistic and that possible consequences should be considered as well. We therefore removed this recommendation and explicitly state that end-users should not

use this strategy but rather that it may be used for model development with further clinical validation and safety testing: “Our analyses suggest that a potential strategy to further increase (self-care) accuracy could involve utilizing output variability, i.e., prompting the model multiple times and aggregating its recommendations. In our controlled setting, selecting the lowest urgency across multiple trials increased self-care accuracy without a large decrease in other urgency levels. However, this approach should not be viewed as a deployment recommendation for end-users, as it could miss emergencies with fatal consequences. Rather, CoT models may apply a pooling algorithm similar to ours to improve the care-seeking advice it provides, with additional clinical validation to ensure that the provided advice is safe for users.”

and

“Because of potential safety concerns, we recommend that this strategy be limited to developers with additional safety testing and not used by end-users. For end-users, it may be reasonable to ask the same question multiple times, and – if possible – across different models (e.g., o4-mini and o4-mini-high, which showed higher self-care accuracy than laypeople) and to critically assess and aggregate the recommendations they receive.”

In the conclusions, we rephrased our recommendation as well: “Prompting different models multiple times and aggregating recommendation can increase their accuracy in identifying self-care cases. Because there is currently no one-size-fits-all model—some models perform better with more urgent cases, while others perform better with low-urgency cases—users should test, critically assess, and aggregate recommendations from multiple ChatGPT models or other non-LLM applications rather than relying on a single general-purpose model.”

4) A point to consider is the discussion around regulation. While I agree that regulation is important when it comes to AI in healthcare, I think the argument needs more nuance. These models aren't being used as medical tools per se. They're being used privately by individuals, often outside any clinical setting. That makes regulation both more complex and, in some ways, harder to enforce. A clearer distinction here would strengthen the paper's argument.

Response: Thanks for this suggestion that we completely agree with. We expanded this section, which now reads:

“This study also demonstrates that AI should be regulated and evaluated according to specific use cases. Because ChatGPT is a general-purpose tool and not marketed as providing medical advice, it is generally not regulated as a medical device.⁴¹ When it is used privately for health information without official medical claims, consumer and data-protection laws apply rather than medical-device rules.^{41,42} Hence, it is important to clarify how future regulation can mandate the safety of ChatGPT and other LLMs, which use cases should be regulated, and whether this should apply only in clinical settings or also to private use.”

5) Some exploration of prompt engineering would be insightful. It would be valuable to see how the different models react to varied prompts or less clearly formulated questions. That would give a better picture of real-world performance, where users don't always phrase things perfectly.

Response: Thank you for the suggestion. Prompt engineering and less clearly formulated questions would indeed be valuable for understanding real-world performance. However, we consider this out of scope of the present work, as we designed this study specifically as a controlled comparison using standardized cases, a fixed prompt, and repeated trials to assess performance in a highly controlled setting. Nevertheless, we added your suggestion as an idea for future research to the Limitations section: “Future studies may also explore different prompting strategies that laypeople use and how they affect the LLMs’ accuracy.”

6) It’s worth mentioning that in a real-world scenario, models can ask follow-up questions to clarify ambiguous inputs. That kind of interactive exchange is an important part of how these tools are used but isn’t covered in this study.

Response: We completely agree with this statement, although it is not completely clear how users would respond to and interact with the LLMs. Nonetheless, this is a clear limitation of our study, so we added the following to our Limitations section: “First, the prompt used in this evaluation is artificial, and laypeople are unlikely to phrase their queries in the same way it was designed for this study. In real-world settings, models may also ask follow-up questions to clarify ambiguous inputs, and users can provide additional information – both of which can affect LLM accuracy.”

7) Some analysis of the errors themselves would be great to better understand the model differences. Are the same cases misclassified across all models?

Response: Thank you for this great idea. We added a heatmap to the manuscript showing the variability of errors between all vignettes. We also added some descriptive information on this figure to the results: “Most errors (69%, 2083/2996, [95% CI 68-71%]) were observed in self-care cases. No self-care case was solved correctly by all models across all trials (0%, 0/13, [95% CI 0-23%]); 10/13 (77%, [50% CI 0-92%]) were solved correctly at least once, and 3/13 (23%, [95% CI 8-50%]) were never solved by any model. Item difficulty (proportions solved for each case) for self-care cases ranged from 0% to 31%. For non-emergency cases, one case (3%, 1/30, [95% CI 0-17%]) was solved correctly by all models in all trials, all (100%, 30/30, [95% CI 88-100%]) were solved correctly at least once, and none remained unsolved (0%, 0/30, [95% CI 0-12%]). Item difficulty ranged from 46% to 100%. Among the two emergency cases, one (50%, 1/2, [95% CI 1-99%]) was solved correctly by all models in all trials; both (100%, 2/2, [95% CI 16-100%]) were solved correctly at least once, and none remained unsolved (0%, 0/2, [95% CI 0-84%]), see Figure 3. Item difficulty ranged from 97% to 100%.”

8) I found the conclusion that users should choose the model based on their case a bit unrealistic. Expecting from users to assess the model’s suitability case by case feels disconnected from how these tools are actually used and how much most of the users know about the underlying technology.

Response: We agree that this conclusion may have been too idealized and too specific to deliver value to end-users. Hence, we rephrased the conclusions to state that users should test multiple models and critically assess and aggregate the recommendations: “Because there is currently no one-size-fits-all model—some models perform better with more urgent cases, while others perform better with low-urgency cases—users should test, critically assess, and aggregate

recommendations from multiple ChatGPT models or other non-LLM applications rather than relying on a single general-purpose model.”

Reviewer #2:

The authors benchmark 16 ChatGPT models on 45 validated care-seeking vignettes, each prompted 10 times (7200 assessments). They report overall accuracy (~70%) remains stable across model generations, with modest improvements in self-care classification among chain-of-thought (CoT, "reasoning") models. They also explore across-trial variability and demonstrate that pooling strategies, particularly selecting the lowest urgency across trials, can increase accuracy, especially for self-care. The study concludes that while ChatGPT models outperform laypeople in some respects, their accuracy remains insufficient for standalone clinical use.

The manuscript addresses an important, practical question and brings several innovations (breadth of model benchmarking, aggregation strategies). However, the numerical inconsistencies, emergency case scarcity, unclear variability narrative, incomplete reproducibility details, and some overstatements of practical implications are significant issues. Also, since GPT-5 is now publicly available, incorporating it into the benchmarking would substantially improve the timeliness and relevance of the manuscript.

Major Strengths

- Breadth and longitudinal perspective: The inclusion of essentially all available ChatGPT models, tested under a consistent protocol, adds valuable longitudinal context beyond single-model snapshots.
- Attention to reproducibility: The study carefully describes prompting procedures, context clearing, and dates of model access, enhancing transparency.
- Focus on variability and aggregation: Evaluating across-trial instability and testing pooling rules is both novel and practical, and contributes to understanding how real-world users might experience LLM advice.
- External validity: Vignette set is previously validated, stratified by base rates, and grounded in real patient cases.

Major Concerns

1. Several accuracy values in text and figures do not match the reported numerators/denominators - like:

- o1-mini...78% (331/450), but $331/450 = 73.6\%$.
- o4-mini...22% (69/130), but $69/130 = 53.1\%$.

However the confidence intervals align with the corrected point estimate, suggesting typographic or calculation errors. Regardless, these need systematic reconciliation before the results can be interpreted with confidence.

Response: Thanks for making us aware of that. This was indeed due to a code error, which has now been fixed. We additionally reviewed all values manually, which now match the numerators/denominators.

2. Only two emergency cases were included. Is that a limitation of the validated set? Regardless, claims such as “all models correctly identified all emergencies” should

be softened, as accuracy estimates and CIs are unreliable with $n=2$. The authors should either expand the emergency case set or temper conclusions about this class.

Response: Including only two emergency cases is indeed a limitation of the dataset used, as it stratified the cases by triage levels and true emergencies are relatively rare. We rephrased all claims about emergency care and toned our conclusions down. For example, the results section now reads that most models “correctly identified both emergency cases” instead of “all emergency cases”. Similarly, in line with your seventh comment, the implications section now does not recommend using ChatGPT for emergency guidance, but rather that “medical laypeople may therefore use them and critically assess recommendations if they are uncertain whether emergency care is necessary, although they are not sufficient for standalone use; when in doubt, users should follow local emergency guidance.”

3. The text claims variability “tends to increase with newer models,” but Table 1 shows excellent kappas for gpt-4.1/4.5-preview (0.96–0.97). Figure 3 suggests more nuanced patterns. The narrative should be aligned with the actual variability measures.

Response: We agree that this trend may not be applicable to all models, as gpt-4.1 and 4.5-preview show very low variability. We rephrased this section accordingly: “The output for the same case varied across multiple trials with no clear trend over time. However, some models (such as gpt-4.1 and gpt-4.5-preview) demonstrated very low variability”. Similarly, in the discussion, we now state: “It should be noted, however, that variability strongly depended on the specific model used”.

4. Variability is highly sensitive to decoding parameters. The manuscript does not report temperature, top_p, max_tokens, or seed values used in OpenRouter calls. Without these, reproducibility is limited. Please include full parameterization and any relevant system prompts.

Response: Thanks for this suggestion. We used the default settings of the OpenRouter API and clarified the exact decoding parameters in the manuscript now.

5. The authors use GPT to classify GPT. Output classification relied in part on gpt-4.1-nano/mini, with human adjudication of disagreements. Because those same model families are under evaluation, correlated errors are possible. A sensitivity analysis using only human labels on a subset would strengthen confidence that findings are not biased by automated adjudication.

Response: We are somewhat surprised by this comment, because we anticipated correlated errors and already classified 50% of all cases manually (by two raters, who re-assessed and discussed any disagreements) and compared the GPT classifications against this manual classification. The models agreed with all manual classifications (which is unsurprising, because the tested models gave a clear recommendation for any of three triage levels).

6. The introduction implies that output variability “has not yet been examined” for care-seeking advice, but prior work by Fraser for WebMD symptom checkers is arguably such an analysis. Levine (16) too. Similarly, define and cite “CoT models”

where first introduced.

Response: Although we agree that variability in outputs more generally has been assessed in other studies on care-seeking advice, output variability in the sense that the same LLM gives differing outputs for exactly the same input was not assessed in these studies. Studies on symptom checkers like WebMD assessed differing output due to differing input, and Levine et al. assessed the output variability of different prompts with different examples. To make that clearer, we rephrased the research gap accordingly: “Although output variability for the same input has been identified as a challenge when LLMs were used by professionals for diagnosing and treating patients,^{6–10} it has not yet been examined in the context of LLMs giving care-seeking advice. Prior studies have examined output variability only in non-LLM symptom-assessment applications or across prompts that use different examples.^{16,26}”

Additionally, we defined CoT models in the introduction: “... chain-of-thought (CoT) models (that are built on top of regular models and are automatically instructed to critically assess and revise their answers before giving output to simulate human meta-cognition)...”.

7. The Discussion suggests laypeople should use ChatGPT to decide about emergencies. Given safety, regulatory, and liability concerns, and the very limited emergency sample, this should be reframed more cautiously, like “may assist but is not sufficient for standalone use; when in doubt, follow local emergency guidance.”

Response: We agree that this section had to be toned down. We followed your suggestion so that it now reads: “Medical laypeople may therefore use them and critically assess recommendations if they are uncertain whether emergency care is necessary, although they are not sufficient for standalone use; when in doubt, users should follow local emergency guidance”

8. Since GPT-5 is now publicly available and many older models are deprecated, the authors should strongly consider incorporating it into their analysis, as it would substantially improve the timeliness and relevance of the manuscript.

Response: We agree that including GPT-5 increases the quality and timeliness of the manuscript. Thus, we collected data from GPT-5 as well and included it in our analysis.

Minor Comments

- Clarify terminology: The text alternates between “laypeople” and “patients” as if disjoint; define terms consistently.

Response: Because ChatGPT is not only used by patients but medical laypeople more generally, we rephrased accordingly and write about “medical laypeople” throughout the manuscript now.

- Abstract and Results numbers must match exactly.

Response: Numbers in the abstract and results now match exactly.

- “In addition to overall accuracy, also output variability...” (smoother phrasing needed)

Response: We rephrased this sentence: “In addition to overall accuracy, output variability of LLMs across multiple trials using the same input should also be examined”

- Low-resolution figures (especially axis labels in Figures 1–4) should be improved.

Response: The figures were included as vector graphics, which were probably converted to low-resolution images in the review portal. They will be included as vector graphics for the final publication.

- Consider placing emergency vignette results in Supplementary material with caveats in the main text.

Response: Thanks for this idea, that we considered. Given that we report average accuracy as the main outcome, we would like to keep emergency vignette results in the Results section, because it influences the average accuracy. The vignettes were stratified to reflect natural base rates of emergency, non-emergency and self-care cases, and true emergencies are relatively rare in the real world. Thus, we think that important information would be missing if we moved the emergency vignette results to supplementary material. However, as per your other comments, we substantially toned down the language when writing about emergency case results.

Statistical Analysis and Reproducibility: The analytic framework (proportions, confusion matrices, Cohen’s kappa, ICCs) is appropriate. However, the numerical inconsistencies undermine confidence in the reported estimates. Reproducibility is limited by omission of key decoding parameters and reliance on LLM adjudication without full human replication.

Response: We carefully assessed numerical inconsistencies, added decoding parameters, and also emphasized the high agreement (100%) with human adjudication in 50% of all cases in the manuscript.